# A Cross-Scale Framework for Modelling Chloride Ions Diffusion in C-S-H: Combined Effects of Slip, Electric Double Layer and Ion Correlation

**DOI:** 10.3390/ma15228253

**Published:** 2022-11-21

**Authors:** Yunchao Qi, Weihong Peng, Wei Zhang, Yawen Jing, Liangyu Hu

**Affiliations:** 1School of Mechanics and Civil Engineering, China University of Mining and Technology, Xuzhou 221116, China; 2State Key Laboratory for Geomechanics and Deep Underground Engineering, China University of Mining and Technology, Xuzhou 221116, China

**Keywords:** cross-scale, interface effect, C-S-H, diffusion

## Abstract

Water and chloride ions within pores of cementitious materials plays a crucial role in the damage processes of cement pastes, particularly in the binding material comprising calcium-silicate-hydrates (C-S-H). The migration mechanism of water and chloride ions restricted in C-S-H nanopores is complicated due to the presence of interfacial effects. The special mechanical properties of the solid–liquid interface determine the importance of boundary slip and Electric Double Layer (EDL) and ion diversity in pore solutions determines the difference of the EDL and the stability of water film slip. A cross-scale model covering slip effects, time-varying of EDL and ion correlation needs to be developed so that the interfacial effects concentrated at the pore scale can be extended to affect the overall diffusivity of C-S-H. The statistics of pore size distribution and fractal dimension were used to quantitatively compare the similarities between model and C-S-H structure, thus proving the reliability of cross-scale reconstructed C-S-H transmission model. The results show that the slip effect is the dominant factor affecting the diffusion ability of C-S-H, the contribution of the slip effect is up to 60% and the contribution rate of EDL time-varying only up to about 15%. Moreover, the slip effect is sensitive to both ion correlation and C-S-H inhomogeneity and EDL time-varying is almost insensitive to ion correlation changes. This quantification provides a necessary benchmark for understanding the destructiveness of cement-based materials in the salt rich environment and provides new insights into improving the durability of concrete by changing the solid–liquid interface on the micro-nanoscale.

## 1. Introduction

Concrete, as the main material for infrastructure, is widely used in oceanographic, hydraulic and highway engineering due to its stable mechanical properties. However, when concrete materials are put into service in a chlorine rich environment, physical and chemical phenomena that affect the structure durability will occur, and major safety hazards and economic losses follow. Consequently, the durability of concrete caused by chloride ions cannot be ignored [1,2]. The erosive medium (water, chloride ions) is transported in concrete through the gel pores in C-S-H, the capillary pores in cement, the interfacial transition zone (ITZ) of aggregate-bulk paste and the cracks. The durability of concrete materials serving in chloride environment largely depends on the transmission rate of chloride ions [3].

Under the background of wide application of high-performance concrete with low water cement ratio, the volume fraction of capillary pore and interface area in concrete is very small. According to percolation theory, capillary pores and ITZ cannot form a connected pore grid, and the pore phase in C-S-H, as an important component of the concrete seepage, becomes an important transmission channel for aggressive media and an important factor affecting the durability of concrete. In addition, C-S-H is one of the main products of cement hydration, accounting for 50~60% of the volume of fully hydrated cement paste. C-S-H plays a decisive role in increasing the strength of concrete mixtures and the mechanical and physical stability of hardened cement pastes [4,5]. Therefore, C-S-H is the most important phase in cement-based cementitious materials and the most important factor affecting the macroscopic behavior of cement-based materials. It is very meaningful to explore the transmission capacity of aggressive media in C-S-H. Many scholars have explored the pore phase C-S-H based on different C-S-H microstructure models and advanced experimental techniques. Bapat [6] considered that more than 90% of the C-S-H pore system is gel pore and contains 6~10% capillary pore. Das and Atzeni [7,8] believe that the internal pore of C-S-H is mostly in the range of 0.5 nm to 20 nm. Lange and Jennings [9] compared the pore size distribution measured by mercury-in-pressure (MIP) with that obtained by backscattering (BSE) images by using image analysis techniques, and concluded that the pores in C-S-H mainly consisted of small-sized (0.5~10 nm) gel pores and large capillary pores (>10 nm). Early scholars thought that C-S-H was homogeneous. Powers thought that C-SH was formed by the accumulation of rigid C-S-H particles with 14 nm particle size and the gel porosity was 28% [10]. With the application of advanced nano-testing technology in the observation and measurement of C-S-H, the high-density C-S-H (HD C-S-H) model and low-density C-S-H (LD C-S-H) model proposed by Jennings and others have been proved to be correct, in which the high-density and low-density C-S-H porosity are 24% and 37% respectively [11]. Zhang [12] constructed the three-dimensional structure of C-S-H gel by two-scale (Macro-Micro) and one-scale accumulation methods, and obtained the size, distribution and average pore size of C-S-H gel. In the C-S-H model based on two-scale stacking method, the diffusion in C-S-H gel is mainly determined by Macro scale. The average pore size of Macro scale in High density (HD) C-S-H gel is about 10 nm, and the average pore size of Macro scale in Low density (LD) C-S-H gel is 25 nm. In the gel model based on one-scale stacking, the average pore size is around 5 nm in HD C-S-H and 10 nm in LD C-S-H. It can be seen that the channels available for the transport of aggressive media in C-S-H are mainly concentrated at the nanometer size.

Solid–liquid interface properties play an important role in influencing the migration of water and chloride ions at micro and nanoscale. At the micro and nanoscale, the liquid between the solid–liquid interface exhibits the typical characteristics of a confined liquid. Confined liquids refer to thin liquid layers that are confined within a very small space and greatly restricted in free movement, and confined liquids often exhibit different mechanical properties and mechanical structures (curing-like, stratification) than bulk liquids. One of the important characteristics of restricted liquids is the stratified structure on the solid surface, which results in density oscillation between the two walls [13]. Wang et al. [14] carried out molecular dynamic simulation of Couette flow and the results show that the water density near the solid wall exhibits oscillatory characteristics, which indicates that stratification occurs in the restricted liquid. In addition to the stratification phenomenon exhibited by confined liquids, boundary slip has also received a lot of attention. Slip is a state in which the volume of crystal changes with respect to another part of liquid along a certain crystal plane and crystal direction under the action of shear stress in the liquid film. The existence of slip effect determines that the slip boundary condition must be used for the nanoflow control equation. The slip boundary condition means that the normal relative velocity component of the liquid on the solid wall is zero, but the tangential relative velocity component is not zero. In addition, the existence of slip boundary conditions leads to the formation of boundary layers in the liquid. Most of the liquid moves above this boundary layer and the velocity of bulk liquid is different from that of the boundary layer. The liquid viscosity and bulk liquid viscosity of the boundary layer are also considered as two different quantities. Strictly speaking, the absolute no slip boundary condition does not exist, because the no slip boundary condition requires that the solid surface and the fluid molecules near it be in an ideal thermodynamic equilibrium state, which is almost impossible [15]. Navier [16] proposed a linear slip model and introduced a linear boundary condition which assumes that the fluid slips directly on the solid surface and that the slip velocity is proportional to the normal component of the fluid strain rate tensor. Slip occurs not only on hydrophilic but also on hydrophobic walls. Vinogradova [17,18,19] observed that shear thinning occurs on the surface of the hydrophobic solid wall and the slip length is related to the viscosity of liquid near the wall and bulk liquid. Based on this, a formula for calculating the slip length of liquid on the surface of hydrophobic solid wall was proposed. To investigate slip in micro and nano flows, Martini [20] proposed a VDFK slip model. The model describes the liquid molecular motion equation in the first layer near the solid wall and points out that the slip length is linearly related to the vertical spacing between the two liquid surfaces. With the development of slip theory, the importance of slip effect in practical engineering is gradually recognized. Song et al. [21] explored the influence of slip effect on fluid motion characteristics in microchannels and calculated the velocity of fluid under slip boundary conditions. On this basis, the velocity equation and the basic differential equation of slip seepage in porous media with slip boundary conditions were derived using capillary bundle model. Han [22] explored the influence of slip effect on moisture transfer in cement-based materials, established a moisture transfer model in a single capillary based on the slip length, and deduced the water permeability coefficient under the slip effect.

As a common interface in nature, solid–liquid interface has a very important influence on its characteristics. Unlike the micro-structure of deionized water at the solid-liquid interface, the free charges or ions in solution are rearranged at the solid–liquid interface to form EDL. The maturity and perfection of the EDL theory has gone through a long process, in which the representative models include Helmholtz model, Gouy-Chapman model and Stern electric double layer model [23,24]. There is no doubt that these models have contributed positively to the knowledge of the EDL structure; however, to a certain extent, they do not accurately reflect the distribution of ions at the solid–liquid interface. Based on the Stern model, Grahame [25] classifies the Stern layer into inner Helmholtz layer (IHP) and outer Helmholtz layer (OHP), which constitutes the classic three-layer theory in the EDL. This improvement can better explain the distribution of charged ions at the solid–liquid interface, and thus promote the application of the EDL model. In cement-based materials, the EDL mainly exists at the interface between the pore wall and the pore solution. Friedmann et al. [26] described the ion transport in cement-based materials at micro scale based on the EDL theory and discussed the phenomenon of diffusion layer overlap. The results show that the overlap of diffusion layer also has a great influence on the ion transfer at macro scale. Nguyen and Amiri [27] point out that, due to the existence of continuous liquid, the EDL has a greater effect on saturated concrete than on unsaturated concrete. Hu et al. [28] discussed the phenomenon of chloride ion concentration in cement-based materials and its relationship with microstructure and characteristics of EDL based on the relationship between chloride ion concentration and Zeta potential. Zhang et al. [12] considered the thickness of the EDL in the gel pore, established the C-S-H accumulation model and made a reasonable prediction of the diffusion coefficient of C-S-H gel. The Poisson-Boltzmann equation can generally predict the ion distribution at the solid–liquid interface and the interaction between the electric double layers, especially in the case of low-valent electrolyte solutions or low surface charge density [29,30]. However, when the counter ion valence in the solution is greater than three, the ions adsorbed on the surface of the pore wall will cause the charge of the pore wall to be inverted [31]. Ion correlation is mainly caused by the electrostatic interaction between ions. When the electrostatic force between ions is much larger than the thermal energy between ions, the ion correlation is particularly outstanding [32,33]. Taking concrete pore solution as an example, unlike monovalent sodium ions (${\mathrm{Na}}^{+}$), potassium ions ($K^{+}$) or divalent calcium ions (${\mathrm{Ca}}^{2+}$), trivalent aluminum ions (${\mathrm{Al}}^{3+}$) form a two-dimensional structure of strongly correlated liquid (SCL) on the surface of the pore wall as a high-valence counter ion different from the electrical properties of the nanopore wall. Compared to the attraction of solid wall surface to the counterions, the cohesive energy of strongly correlated liquids will result in additional adsorption between high-valent counterions, and this cohesive energy attracts more counterions to accumulate on the solid wall surface, resulting in the inversion of the solid wall surface charge. This phenomenon needs to be analyzed by strong ion correlation and the Poisson-Boltzmann equation.

Hence, one can see that that the interfacial behavior occurring at the pore scale will have an important influence on the durability of cement pastes, particularly in the binding material comprising calcium-silicate-hydrates. In order to deeply understand the transport behavior of water and ions in nanopores, scholars have carried out many related research studies. Hou [34] modified the slip length, dynamic contact angle and viscosity of the restricted liquid to make the traditional Lucas-Washburn equation more suitable for describing the water migration characteristics at the nanoscale. In addition, Hou established the coarse-grained MD model by molecular dynamics method and compared the results of molecular dynamics with those of a modified LW equation, which indicated the reliability of molecular dynamics in describing the micro-interaction between molecules. In Zhou’s study [35], the interaction between chloride solution and tobermorite substrate was simulated by molecular dynamics. The simulation results show that the presence of calcium ion improves the amount of chloride adsorption and the stability of chloride adsorption. By synthesizing C-S-H and immersing it in chloride solution, Zhou proved that calcium ion promoted the improvement of C-S-H chloride adsorption capacity. Wang [36] quantitatively analyzed the influence of hydration on C-S-H surface based on molecular dynamics theory and density functional theory. The results showed that lubrication between C-S-H interfaces was negatively correlated with water content, and the lubrication mechanism was significantly different between high water content system and low water content system. This research explains the origin of interfacial micro-lubrication at atomic scale and provides a new understanding for the workability design of fresh concrete. Yu [37] proposed the idea of modifying the inner surface of nanopores with composite coating of graphene oxide and epoxy resin based on the fact that adding graphene into cement-based materials can inhibit the entry of aggressive ions. The results show that the presence of composite coating not only limits the migration rate of sodium and chloride ions, but also separates ions from solution, thus restricting the passage of sodium and chloride ions through gel pore. Yang [38] simulated capillary transport of NaCl solution in different size nanopore based on molecular dynamics. The research shows that, with the decrease of pore size, sodium and chloride ions will be forced to separate from water molecules and remain at the entrance of gel pore. In addition, the surface of C-S-H is easier to immobilize sodium and chloride ions than that of water molecules.

It can be seen that the special mechanical properties of the solid–liquid interface determine the importance of boundary slip and EDL. Ion diversity in pore solutions determines the difference of the EDL and the stability of water film slip. The presence at pore-scale of interfacial effects limits the water and chloride ion migration, thus affecting the diffusion capacity of C-S-H. A cross-scale model covering slip effects, time-varying of EDL, and ion correlation needs to be developed so that the interfacial effects concentrated at the pore scale can be extended to affect the overall diffusivity of C-S-H. In this research, the micron-scale C-S-H model was reconstructed based on the C-S-H nucleation growth theory combined with the four-parameter (QSGS) method of random growth, and the surface fractional dimension was used to verify the rationality of the reconstructed model. Based on the diffusion fluxes of individual pores and C-S-H inhomogeneity, the diffusion coefficients of C-S-H were modified, and finally a non-homogeneous diffusion model at the micron scale was established. The effects of inhomogeneity, slip effect, bilayer time variability, and ion correlation on the overall diffusion ability of C-S-H were investigated comprehensively.

## 2. Experimental and Computational Method

### 2.1. Establishment of Migration Model at Pore Scale

In our previous studies [39], we established a transport model of chloride ion and water considering interfacial effects at pore scale. Figure 1a,b show the distribution of water and ions at the solid–liquid interface under weak and strong ion correlation, respectively. The distribution of ions in the electric double layer under weak ion correlation can be described by the Poisson-Boltzmann equation, which will be broken in the case of strong ion correlation. High valence counterions from pore solutions forcibly displace low valence counterions in compact layer and combine four water molecules to form a strongly correlated liquid with two-dimensional structure. The cohesion of strong correlated liquids in solution will cause additional adsorption between high valence counter ions, which will attract more counter ions to the solid wall surface than the solid wall surface itself. Therefore, the charge on the solid wall surface will be inverted to form a new double layer structure. At this point, the counter ions no longer coexist with the co-ions in the diffusion layer, and the slip effect at the solid–liquid interface no longer exists. 

The length of the model is set as 100 nm, the basic aperture is set as 10 nm, wedge coefficient is set as 1.0. The left and right boundaries of the model are the inlet and outlet of fluid transport, respectively, and the chloride concentration at the inlet is 0.44 mol/L, which is close to the upper limit of chloride concentration of pore solution in cement-based materials. The initial thickness of EDL is 2.038 nm, the thickness of the compact layer is 1.068 nm and the thickness of the diffusion layer is 0.97 nm. The time-varying velocity of diffusion layer is determined by the Poisson-Boltzmann equation and the Einstein-Stokes equation, the time-varying process of the electric double layer is simulated by the dynamic grid module in COMSOL Multiphysics. More detailed information about the governing equations and boundary conditions can be obtained from our previous research.

### 2.2. Reconstruction of C-S-H Model

The basic principle of the four-parameter stochastic growth method is proposed in combination with the pore growth model and clumping mechanism of porous media configuration [40]. In this study the method is used to reconstruct the C-S-H model, and the process of reconstructing C-S-H by this method has a high similarity with the C-S-H nucleation growth theory. For two-dimensional C-S-H model, solid cement particles can be selected as growth phase, pore is non-growth phase and initial distribution is full of pore. We consider that only single growth phase of gel particles exists in the C-S-H model, and the first three parameters are mainly set during the structure simulation process: initial growth nucleus distribution probability *P_d_*, growth probability *P_m_*, volume fraction expectation *P_v_*. *P_v_
* determines the proportion of solid particles in the model, reflecting the sparse and dense distribution of particles in porous media, the larger *P_v_*, the fewer gaps in the model. *P_d_* represents the probability that a blank grid in a given area will become a growth core. When *P_d_* is small, the internal micromorphological characteristics of porous media including the skeletal solid particle size, the shape of the pores and the connectivity structure between the skeleton and the pores are described more finely. When *P_d_* is large, the solid phase particles are described more coarsely and the skeletal phase tends to be uniformly distributed in the media and the average void space between the skeletons is small. *P_m_* indicates the likelihood that the nucleus will grow in the surrounding direction. The smaller the *P_m_*, the more accurate the final model porosity will be, but the more iterations will occur and the more computing resources will be consumed. As shown in Figure 2 and Figure 3, the model was constructed. 

As shown in Figure 4, we show the results of the above algorithm by generating C-S-H models with 24%, 26%, 28% and 30% porosity, respectively; the black part represents the gel growth phase and the white part represents the pore phase. The corresponding volume fraction expectation *P_v_* are 76%, 74%, 72% and 70%, respectively, *P_d_* was set 0.0001 and *P_m_* was set 0.001 during process. Current research shows that the total porosity of C-S-H is about 28%. Therefore, in subsequent studies we set a porosity of 28%. More work was done to discuss the rationality of parameter setting of *P_d_* and *P_m_*.

### 2.3. Rationality Verification of QSGS Model

In this paper, a QSGS model was used to simulate the C-S-H structure to analyze the physicochemical and seepage properties of C-S-H. However, whether the model can represent the C-S-H structure still lacks quantitative research. To study the similarity between QSGS model and C-S-H structure, so as to provide a scientific basis for parameter selection when applying the model, this section evaluates the similarity between QSGS and C-S-H structure by comparing the pore distribution and fractal dimension values of C-S-H and QSGS model. 

#### 2.3.1. Pore Distribution

Quantifying migration channels is not only a data source for comparing QSGS models with C-S-H structures, but also a basis for quantifying interface effects and their changes. In this study, Field Emission Scanning Electron Microscopy (FE-SEM) and Atomic Force Microscopy (AFM) were used to observe the pore structure in C-S-H, providing parameters for the establishment of physical model.

AFM from Bruker Nano Lnc, Germany, with a maximum scanning range of 90 µm in XY direction and 10 µm in Z direction; Low drift level, 0.2 nm/min/°C; Resolution, Atomic resolution (calibrated with mica, 0.27 nm transversely and 0.05 nm high) can be obtained continuously and steadily. Cement paste samples were prepared with a water-cement ratio of 0.35 and an additive content of 2%. Ordinary Portland cement is used with a density of 3.15 g/cm^3^ and the additive uses Polycarboxylic Water Reducer with a solid content of 30% and a water reducing rate of 35%. The specimens were vacuum-dried after 28 days of standard maintenance. Cement paste test block was finally made into slices of 15 mm × 15 mm × 5 mm with surface undulations not exceeding 15μm. In this study, argon ion polishing is used to ensure the surface smoothness of the sample and the slice height is 100μm higher than the baffle plate when it is fixed, in order to obtain the true pore cross-section morphology inside the C-S-H and avoid the influence of pore shadows. The high-voltage electric field is used to make argon produce ionic state. The argon ion generated will bombard the surface of the sample at high speed under the acceleration voltage and peel off the surface layer by layer to achieve the polishing effect. The section was cleaned with anhydrous ethanol before observation to remove surface-adsorbed impurities. As shown in Table 1, the observed nanopores were statistically and comprehensively processed.

Cement based material samples were cut into several pieces of 15 mm × 15 mm thin sections. To obtain high-resolution imaging and prevent the accumulation of electrostatic field on the sample caused by electron beam scanning during the test from affecting the test results, the sample was sprayed with gold before the test. The samples of cement paste were analyzed by FE-SEM (TESCAN MAIA3). The main performance indexes are as follows: the electron gun emits the electron gun for high brightness field Schottky field, amplification times are 4–1,000,000, the electron beam energy is 200~30 keV, the electron beam current is 2 pA~400 nA, the maximum field of vision is 4.3~7.5 nm. The high voltage (SEM HV) is 2.0 kV. The type of the detector is SE. As shown in Table 2, the observed nanopores by FE-SEM were statistically and comprehensively processed.

It can be seen from the research results that the pore size in C-S-H pore system mainly concentrates on 10 nm to 30 nm. To verify the reliability of the QSGS model, the pore distribution in the QSGS model was compared with the results of quantitative analysis by AFM and FE-SEM. QSGS models are formed by combining different types of parameters and their values. Considering that the calculation results of the same type of combination have the same or similar rules and limited to the length of the article, this research takes the representative combination as an example to give the corresponding calculation results and discuss them.

QSGS model was reconstructed with 28% porosity, *P_d_
*= 0.001 growth nucleus distribution probability and 28% porosity, growth probability *P_m_
*= 0.001, respectively. Pores in QSGS model with different parameters and pores in C-S-H were divided into 10~12 pore size classes within the pore size range, then the proportion of pores smaller than this class in the total number of pores is counted.

As shown in Figure 5a, the trend of pore size distribution curve in QSGS models with different growth probability *P_m_* values is generally consistent with that in C-S-H, but slightly difference in small pore size distribution. It is not difficult to find that both pore size distributions are concentrated in the range of 10 nm to 40 nm, while the pore size distribution range in the QSGS model is generally small. The similarity of pore size distribution between QSGS model and real C-S-H structure has no significant correlation with the *P_m_* value. This can be explained by the fact that the *P_m_* indicates the possibility of the growth core growing in the surrounding direction, and the value of *P_m_* determines the accuracy of the porosity in the QSGS model, but has no obvious effect on the pore distribution in the model. Therefore, in order to obtain more accurate porosity and considering the number of iterations, the value of *P_m_* was set as 0.001. As can be seen in Figure 5b, the pore size distribution curve of the model is consistent when *P_d_
*= 0.0001~0.005, and it is in good agreement with the pore size distribution curve of C-S-H. However, the pore size distribution curve of *P_d_
*= 0.01~*P_d_
*= 0.1 model gradually deviates from that of C-S-H and *P_d_* = 0.0001~0.005 model, and the deviation trend increases with the increase of growth core distribution probability. In addition, it can be seen that the pore distribution in C-S-H model reconstructed by QSGS based on *P_d_
*= 0.0001 and *P_m_
*= 0.001 is consistent with that in the real C-S-H model.

#### 2.3.2. Surface Fractal Dimension

Fractal phenomenon is an important index to describe pore structure of porous media, and its fundamental property is self-similarity, which means that there is a certain similar shape both locally and globally [41,42]. Pore morphology of C-S-H structure also has certain self-similarity and scale invariance, so fractal method can be used to study pore structure of C-S-H [43]. Fractal dimension can be used to characterize the sectional structure of porous system and to quantitatively analyze and compare the QSGS model with the real C-S-H pore structure. Pore surface fractal dimension *D_S_* is used to represent the roughness of the pore boundary. 

In this research, based on the FE-SEM experimental results, the distribution characteristics of C-S-H surface pores are quantitatively analyzed by Box-counting Dimension [44]. Figure 6 shows the process of calculating the fractal dimension of pore with box dimension. First, based on the original SEM image, select the appropriate threshold to convert the gray image into black-and-white binary image, then select different boxes with different edges $\left(\delta \right)$ to cover the area of the black-and-white binary image, and finally count the number of small boxes containing pore. Using the following formula, the fractal dimension can be calculated
(1)$$D=-\underset{\delta \to 0}{\lim}\frac{\ln(N(\delta ))}{\ln(\delta )}$$

In general, the surface fractal dimension *D_S_* value reflects the degree of curvature of the pore profile. The rougher the pore contour boundary and the more irregular the shape, the larger the surface fractal dimension *D_S_*.

In the above part, we compared the pore quantity distribution curves between the QSGS model generated according to parameters volume fraction expectation *P_v_*, *P_d_* and *P_m_* were 72%, 0.0001 and 0.001, respectively. The pore quantity distribution curves of the QSGS model agree well with those of C-S-H. In this section, we compare the fractal dimensions of the pore surface in the QSGS model under this parameter with those in the C-S-H model. As shown in Figure 7, the surface fractal dimension *D_S_* of QSGS model ranges from 1.25 to 1.55, which is very close to the range of surface fractal dimension *D_S_* of pore in FE-SEM image. The surface fractal dimension *D_S_* of FE-SEM image is 1.21 to 1.51, which indicates that the QSGS model is consistent with the roughness and bending degree of actual C-S-H pore contour boundary. In addition, the distribution of surface fractal dimension was fitted and the results show that the distribution of pore surface fractal dimension in the model is similar to that in C-S-H.

#### 2.3.3. The Migration Model of Chloride Ions in C-S-H

In previous studies, we discussed the influence of slippage, time-varying of EDL and ion correlation on migration of chloride ion, and the results were presented in the form of molar flux of chloride ion. In order to extend the interfacial effect at pore scale to the influence on the overall performance of C-S-H, we consider C-S-H as a bundle of tubular capillaries with different pore diameters and assume that all solutions passing through the pore are perpendicular to the cross section, then the total flow through C-S-H is the sum of the flows through all single pores. By introducing the chloride ion flux in a single pore under the interfacial effect into the reconstructed C-S-H diffusion model, the chloride ion diffusion flux in C-S-H under the interfacial effect can be obtained. Assuming the cross-sectional area of the nanopore is *A*(*r*), the flow rate at the outlet of a single nanopore is calculated from the following equation
(2)$$q_{I}=u_{\mathrm{avg}(r)}\cdot \rho_{c}\cdot A(r)$$
where $u_{\mathrm{avg}(r)}$ is the average flow rate along the pore length at the outlet of a single nanopore (obtained by dividing the flow line integral at the outlet of the nanopore by the effective pore size of the pore in our previous study [39]), m/s. 

In C-S-H gels, assuming a nanopore beam model with an effective pore radius $r_{c}=h/2-h_{w}$, the total flux can be obtained by integrating all the nanopores.
(3)$$J_{I}=\int_{0}^{r_{c}}u_{\mathrm{avg}(r)}\cdot \rho_{c}d\Omega_{A}$$
where $\Omega_{A}$ represents that the following pore distribution function is satisfied on any section perpendicular to the flow direction [22].
(4)$$d\Omega_{A}=\frac{dV}{\varsigma }\phi_{I}$$
where V is the normalized volume and ς is the wedge coefficient.

The average diffusion coefficient of C-S-H can be concluded
(5)$$D_{I}=\frac{L\cdot \phi_{I}}{\Delta C\cdot \varsigma }\cdot \rho_{c}\int_{0}^{r_{c}}u_{\mathrm{avg}(r)}dV$$
where ΔC represents Chloride ion concentration difference.

C-S-H is not homogeneous and contains LD C-S-H and HD C-S-H. The structure of LD C-S-H is relatively loose while HD C-S-H is relatively dense. The diffusion coefficients of LD C-S-H and HD C-S-H are different due to their different structures. Therefore, the relative proportion of LD C-S-H and HD C-S-H in C-S-H has a very important influence on the diffusion coefficients of C-S-H. C-S-H is the main product of cement hydration, and the volume fraction of each phase in the hydration product is affected by degree of hydration and cement ratio. Powers [10] believes that C-S-H gels are formed by the accumulation of rigid C-S-H gel particles with a particle size of 14 nm and the porosity between the gel particles is 28%. The J-T (Jennings-Tennis) model assumes that the porosity of HD and LD C-S-H is 24% and 37%, respectively [11]. In the J-T model, nitrogen adsorption experimental statistical regression was used to determine the mass ratio of LD C-S-H and HD C-S-H. The volume fraction of HD C-S-H and LD C-S-H can be expressed as:(6)$$V_{HD}=\frac{M_{t}-(M_{r}M_{t})}{\rho_{HD}}$$
(7)$$V_{LD}=\frac{M_{r}M_{t}}{\rho_{LD}}$$
(8)$$M_{r}=3.017(w/c)\alpha -1.347w/c+0.538$$ where *V_HD_* is volume fraction of HD C-S-H, *V_LD_* is volume fraction of LD C-S-H, *M_r_* is the mass fraction of C-S-H that nitrogen can enter, *M_t_* is the total mass of C-S-H gel, *w*/*c* is water-cement ratio and α represents degree of hydration.

It can be seen from the above equation that the volume fractions of HD C-S-H and LD C-S-H depend on water-cement ratio and degree of hydration. Therefore, there is no obvious boundary between LD C-S-H and HD C-S-H, only their relative proportion. With the decrease of water-cement ratio and the increase of hydration time, the proportion of HD C-S-H increases continuously and the proportion of LD C-S-H decreases gradually. Compared with LD C-S-H, HD C-S-H has a more compact structure and smaller internal pores, which also means the interface effect is more obvious. With wider application of high-performance concrete with low water-cement ratio in practical engineering, the proportion of HD C-S-H gradually increases and the influence of interface effect also plays increasingly important role in the influence of overall diffusion properties of C-S-H. Therefore, the interface effect cannot be ignored when exploring the diffusion ability of C-S-H. In this research, parameter *P_r_* which indicates the proportion of HD C-S-H to the total C-S-H was introduced to further explain the influence of water cement ratio and hydration time on interface effect. At high water–cement ratio, the *P_r_* value is about 20%. At low water–cement ratio, the *P_r_* value can reach about 80% [11]. As shown in Table 3, this chapter discusses four cases where HD C-S-H accounts for 20%, 40%, 60% and 80%, respectively. The model size is 3 μm × 3 μm. As shown in Figure 8, according to the algorithm in the previous section, the obtained model is imported into COMSOL Multiphysics. The chloride ion transport process is controlled by dilute matter transfer module. The left and right boundaries are no flux. The lower boundary is the inlet of chloride ion solution. The boundary condition at the inlet adopted the concentration boundary condition and the concentration value was set as 0.44 mol/L, which is consistent with the inlet boundary condition of the model at pore scale. The upper boundary is the outlet of the chloride ion solution. The boundary condition at the outlet still adopted the concentration boundary condition and the concentration value was set as 0 mol/L, which is consistent with the outlet boundary condition of the model at pore scale. 

## 3. Results and Discussion

### 3.1. Migration Model Analysis on Pore Scale

In our previous studies, we discussed the effects of slippage, EDL time-varying and ion correlation on chloride ion migration at pore-scale. In order to briefly exhibition the influence of interfacial effect on chloride ion migration at pore-scale, we showed the molar flux of chloride ions in different pore size at steady state, as seen in Figure 9. Based on the results of pore scale studies and the reconstructed C-S-H model, a cross-scale model covering slip effects, EDL time-varying and ion correlation is established, and the interfacial effects concentrated on the pore scale are extended to affect the overall diffusivity of C-S-H.

### 3.2. The Effect of C-S-H Anisotropy on Diffusion Flux

To investigate the influence of HD C-S-H ratio on diffusion flux, the water and chloride ion migration fluxes in the pore-scale model with pressure gradient of 1.0 Pa, wedge coefficient of 1.0, and strong ion correlation effect under consideration of slip effect and time variability of electric double layer were introduced into the micro scale model. The change trend of water and chloride ion diffusion flux is relatively similar. Therefore, the chloride ion was only taken as an example, as shown in Figure 10. With the increase of $P_{r}$, the proportion of HD C-S-H in C-S-H, the diffusion flux of chloride ions decreases. This is because HD C-S-H has compact structure and smaller pore size. The larger the proportion of HD C-S-H, the smaller the C-S-H diffusion coefficient. When $P_{r}\;=\;0.2$, the diffusion flux of chloride ion in stable state is $7{.74\;\times \;10}^{-4}\;\mathrm{mol}/m\cdot s$, which is 1.05, 1.25 and 1.61 times of that when $P_{r}$ is 0.4, 0.6 and 0.8, respectively. The change trend of water diffusion flux is similar to that of chloride ion. Water diffusion flux in stable state under different parameters is shown in Figure 11. The heterogeneity of C-S-H also has a great impact on the water diffusion flux. When the proportion of HD C-S-H gel increases from 20% to 80%, the water diffusion flux also decreases sharply from $9{.78\;\times \;10}^{-2}\;\mathrm{mol}/m\cdot s$ to $6{.12\;\times \;10}^{-2}\;\mathrm{mol}/m\cdot s$.

### 3.3. The Effect of Slippage on Diffusion Flux 

As shown in Table 4, with consideration of C-S-H heterogeneity, the influence of ion correlation and slip effect on diffusion flux at micron scale was explored. As shown in Figure 12, the chloride ion diffusion flux considering slip effect reaches the stable state more quickly, and the steady state chloride ion diffusion flux considering slip effect is greater than when ignoring slip effect. Under the same conditions, the chloride ion diffusion flux decreases with the increase of $P_{r}$. In case 1, case 2, case 3 and case 4, when $P_{r}$ is 0.2, the proportion of HD C-S-H is 1.60 times, 2.02 times, 1.66 times and 2.27 times that when $P_{r}$ is 0.8, respectively. The diffusion flux under strong ion correlation is generally greater than that under weak ion correlation, and the ion correlation is more obvious under the condition of no slip boundary. Under the effect of slip, the increasing extent of ion correlation on diffusion flux increases with the increase of $P_{r}$, but it is limited. When $P_{r}\;=\;0.8$, it is only 5.96%. Without considering the slip, the effect of ion correlation is more obvious. When $P_{r}$ is 0.2, 0.4, 0.6 and 0.8, the increase of ion correlation is about 14.58%, 15.65%, 19.31% and 28.74%, respectively. This is because the thickness of the electric double layer is small under strong correlation. With the increase of $P_{r}$, the proportion of HD C-S-H and small pores increases. Consequently, the influence becomes larger gradually.

Figure 13 describes the stable state diffusion flux of chloride ion and water with different intensity of ion correlation under the condition of neglecting slip effect and considering slip effect. Under the condition of strong ion correlation, when $P_{r}\;=\;0.2$, the chloride ion diffusion flux neglecting slip only accounts for 62.48% of the flux consideration slip, and this proportion gradually decreases with the increase of $P_{r}$. When $P_{r}$ = 0.4, 0.6 and 0.8, the proportion reaches 60.99%, 57.06% and 49.46%, respectively, which means that the chloride diffusion flux considering slip is more than twice that when the slip effect is ignored. This is mainly because the slip effect is more prominent in smaller nano pores. With the increase of $P_{r}$, the proportion of HD C-S-H gradually increases. Meanwhile, HD C-S-H is compact in structure and contains small pore size.

Compared with the strong ion correlation condition, the slip effect under the weak ion correlation condition is more prominent. With the increase of $P_{r}$, the proportion of chloride diffusion flux when slip is ignored to the flux when slip is considered gradually decreases. When $P_{r}\;$= 0.2, 0.4, 0.6 and 0.8, the chloride ion diffusion flux ignoring slip only accounts for 55.62%, 54.04%, 49.57% and 40.71% of the diffusion flux considering slip, respectively.

### 3.4. The Effect of Time-Varying EDL on Diffusion Flux 

As shown in Table 5, based on the C-S-H heterogeneity, the influence of ion correlation and electric double layer time variability on the diffusion flux in the micrometer scale has been explored. Figure 14 describes the influence of the electric double layer time variability on the chloride ion diffusion flux under different ion correlations when $P_{r}$ (the proportion of HD C-S-H to C-S-H) is 0.2, 0.4, 0.6 and 0.8. It can be clearly seen from the figures that both the time variability of electric double layer and the strong ion correlation increase the diffusion flux of chloride ions. Compared with the case of neglecting time variation of electric double layer, the chloride diffusion flux reaches a stable state faster when the time variation of the electric double layer is considered.

The diffusion flux of chloride ion has little difference under different intensity of ion correlation, and the diffusion flux under strong ion correlation is 1.02 to 1.06 times of that under weak ion correlation. In contrast, the influence of time variation of electric double layer is more prominent. With the increase of $P_{r}$, the influence of electric double layer time variation increases. Under strong ion correlation, when $P_{r}$ is 0.2, 0.4, 0.6 and 0.8, the chloride ion diffusion flux neglecting the time variation of electric double layer accounted for 90.04%, 89.40%, 87.77% and 84.71% of the chloride diffusion flux considering time variation of electric double layer. Under weak ion correlation, the chloride ion diffusion flux neglecting the time variation of electric double layer accounted for 90.18%, 89.60%, 88.12% and 85.25% of the chloride diffusion flux considering time variation of electric double layer, respectively. Different from the phenomenon that the slip effect is weakened under the strong ion correlation, the time variation of electric double layer has little effect on the diffusion flux under different ion correlation. However, the influence of time variation of electric double layer also increases with the increase of $P_{r}$. This is because, the smaller the pore, the more obvious the effect of electric double layer time variation.

Figure 15 shows the diffusion flux when chloride ion and water reach a stable state under different conditions. Taking chlorine ion flux as diffusion an example, the chlorine ion diffusion flux decreases gradually with the increase of $P_{r}$ under the same conditions. In case 1, the chloride ion diffusion flux at $P_{r}\;$= 0.2 is 1.05, 1.25 and 1.60 times of that at $P_{r}\;$= 0.4, 0.6 and 0.8, respectively. In case 2, the chloride ion diffusion flux at $P_{r}\;$= 0.2 is 1.06, 1.28 and 1.70 times of that at $P_{r}\;$= 0.4, 0.6 and 0.8. In case 3, it is 1.06 times, 1.27 times and 1.66 times. Similarly, in case 4, it is 1.07 times, 1.30 times and 1.76 times.

From the above discussion of the research results, it can be seen that the heterogeneity of C-S-H has an influence on the interfacial effect. In order to quantify further the variation of different interface effects with the change of C-S-H homogeneity, the concept of contribution ratio *C* was introduced. The molar flux ignoring the interface effects were set as $F_{i}$ and the flux considering the interface effects were set as $F_{c}$. 

The molar flux considering all the factors, ignoring slippage and ignoring EDL time-varying were set as $F_{ca}$, $F_{is}$, $F_{iE}$, respectively. 

The contribution ratio of slippage ($C_{s}$) can be expressed as
(9)$$C_{s}=\frac{F_{ca}-F_{is}}{F_{ca}}$$

The contribution ratio of EDL time-varying ($C_{E}$) can be expressed as
(10)$$C_{E}=\frac{F_{ca}-F_{iE}}{F_{ca}}$$

Figure 16 illustrates the interaction between interfacial effects and inhomogeneity and their effect on the overall diffusion flux of C-S-H. With the increase of $P_{r}$, the contribution rate of slip effect under strong ion correlation increases from 37.52% to 50.54% and the contribution rate of slip effect under weak ion correlation increases from 44.38% to 59.29%. The existence of strong ion correlation weakens the slip effect, while the increase of $P_{r}$ enhances the slip effect. This is due to the strong ionic correlation at the pore size which increases the adhesive force at the solid–liquid interface and thus weakens the slip effect. The increase of $P_{r}$ value means that the proportion of HD C-S-H increases, the pore distribution tends to be small pore size, and slip effect is more obvious in small pore size. Although the influence of EDL time-varying is more significant in small pore size, the influence of $P_{r}$ value on time-varying EDL is limited. With the increase of $P_{r}$, the increase of contribution rate is only about 5%, and there is no obvious difference between different ion correlations. 

In order to verify the rationality of this study, it is compared with the research of previous research results. It is very difficult to directly measure the diffusion coefficient of chloride ion in C-S-H. So, at present, it is mainly calculated by numerical simulation and inversion method based on the test results of tritiated water transport in hardened slurry. Bentz and Garbociz [3,45] proposed that the relative diffusivity in C-S-H was 0.0025~0.0033 based on the CEMHYD3D model of simulating cement hydration and microstructural development. In this paper, the chloride ion diffusion considering slip and time variation of electric double layer in micro scale is studied. If the influence of pressure gradient and convection is ignored, the relative diffusion coefficient of chloride ion in C-S-H at steady state is between 2.56$\;{\times \;10}^{-3}\;\mathrm{and}\;$4.12$\;{\times \;10}^{-3}$ when $P_{r}\;$= 0.2 to 0.8. The existence of the interface effect promotes the diffusion of water and ions, so it is reasonable that the results of this study are larger.

Therefore, it is of great significance to explore the transport mechanism of chloride ions at micro and nano scale based on slip, time variation of EDL and ion correlation. The results of this study can not only provide ideas for forming the transport mechanism of chloride ions, but also provide a theoretical basis for the research and application of concrete durability.

## 4. Conclusions

The research reconstructs the C-S-H transmission model based on C-S-H nucleation growth theory and combined with QSGS method. The statistics of pore size distribution and fractal dimension were used to quantitatively compare the similarities between QSGS model and C-S-H structure, thus proving the reliability of reconstructed C-S-H transmission model. Interfacial effects occurring at the pore-scale were extended to the C-S-H entirety by a cross-scale model. On this basis, the effects of C-S-H inhomogeneity, slip effect, time-varying of EDL and ion correlation on the diffusion flux of chloride ion were comprehensively discussed, and a prediction model of chloride ion diffusion in C-S-H was established. The main conclusions are as follows:C-S-H is inhomogeneous and the diffusion flux of chloride ion is affected by the relative proportion of HD C-S-H and LD C-S-H. With the increase of the ratio of HD C-S-H, the diffusion flux of chloride ion decreases gradually, because the transmission of chloride ion is hindered due to the compact structure of HD C-S-H.Slip effect is the dominant factor affecting the diffusion ability of C-S-H. With increasing $P_{r}$, the contribution of the slip effect to the diffusion flux is up to 50% under strong ion correlation and this value is even higher, up to 60%, under weak ion correlation.Compared with the contribution rate of slip effect, the contribution rate of EDL time-varying is slightly insufficient and has no significant change under different ion correlations, the contribution rate only up to about 15%.The slip effect is sensitive to both ion correlation and C-S-H inhomogeneity. The contribution rate of ion correlation to the slip effect is about 10%, and the $P_{r}$ value changes from 0.2 to 0.8, resulting in a contribution rate change of about 15%. EDL time-varying is almost insensitive to ion correlation changes; the $P_{r}$ value changes from 0.2 to 0.8, resulting in a contribution rate change of about 5%. It is not difficult to see that the slip effect under the weak ion correlation phase has the strongest reaction to the change of C-S-H homogeneity, and the time-varying reaction of EDL under the strong ion correlation is the weakest.The significance of this study is not only to quantify the influence of interface effects on the overall diffusivity of C-S-H, thus improving the accuracy of predicting the life of concrete, but also to provide new insights into improving the durability of concrete by changing the solid–liquid interface on the micro-nanoscale.

## Figures and Tables

**Figure 1 materials-15-08253-f001:**
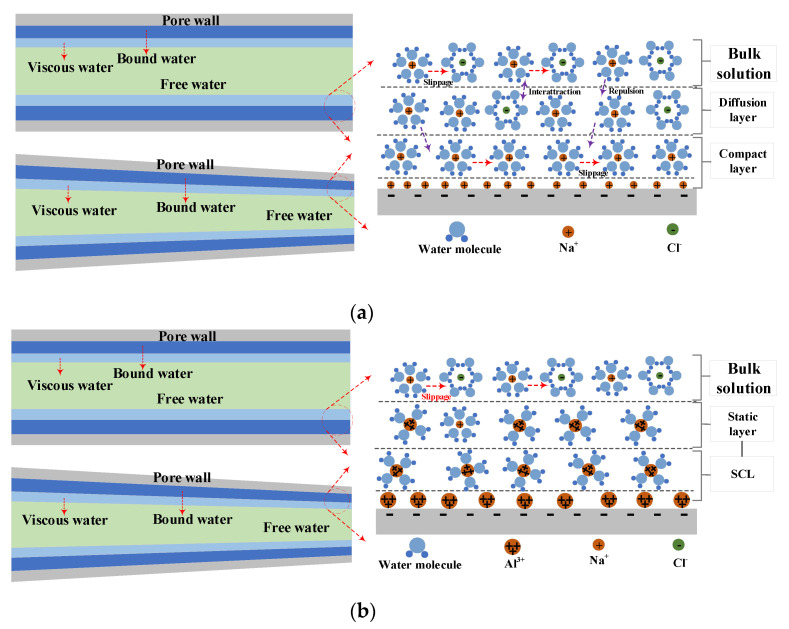
Pore-scale migration model. (**a**). Transport of water and chloride ions under weak correlation. (**b**). Transport of water and chloride ions under strong correlation.

**Figure 2 materials-15-08253-f002:**
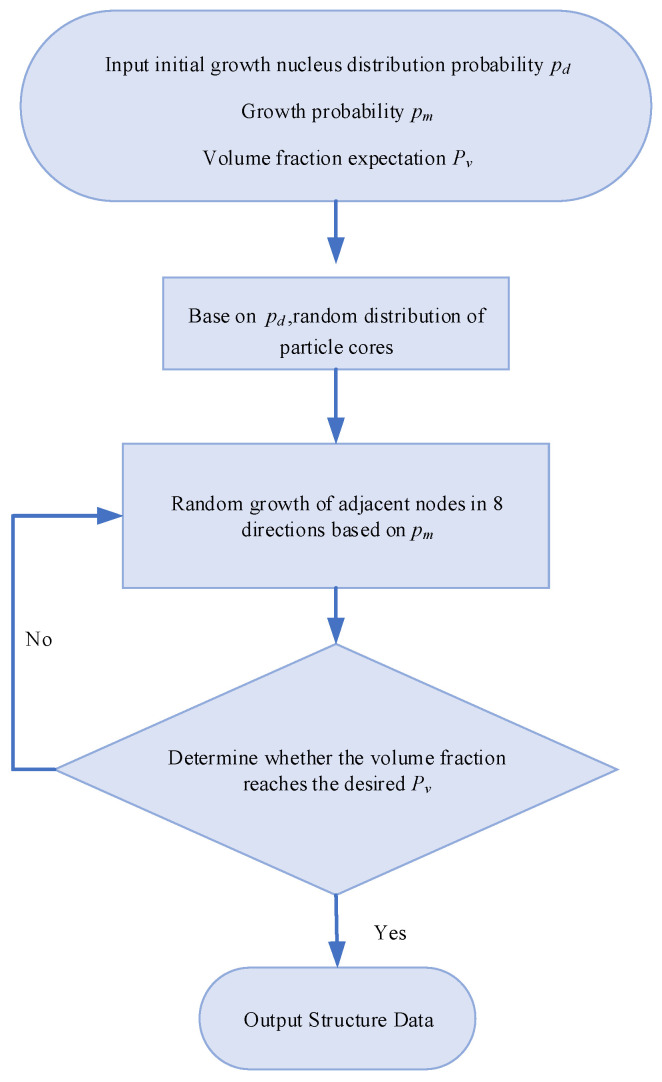
The flow chart of C-S-H nucleation growth.

**Figure 3 materials-15-08253-f003:**
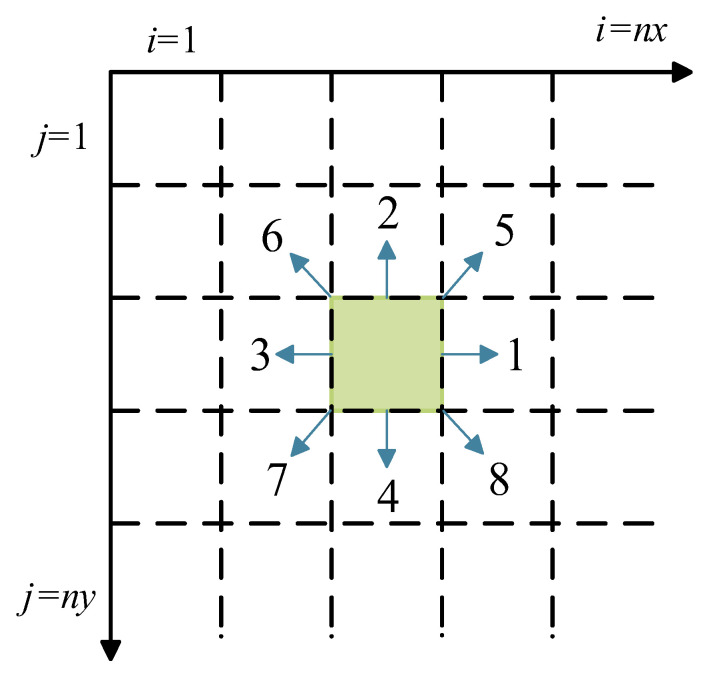
Schematic diagram of C-S-H nucleation growth direction.

**Figure 4 materials-15-08253-f004:**
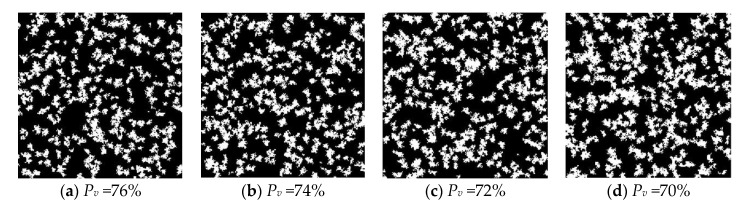
C-S-H models with different porosity.

**Figure 5 materials-15-08253-f005:**
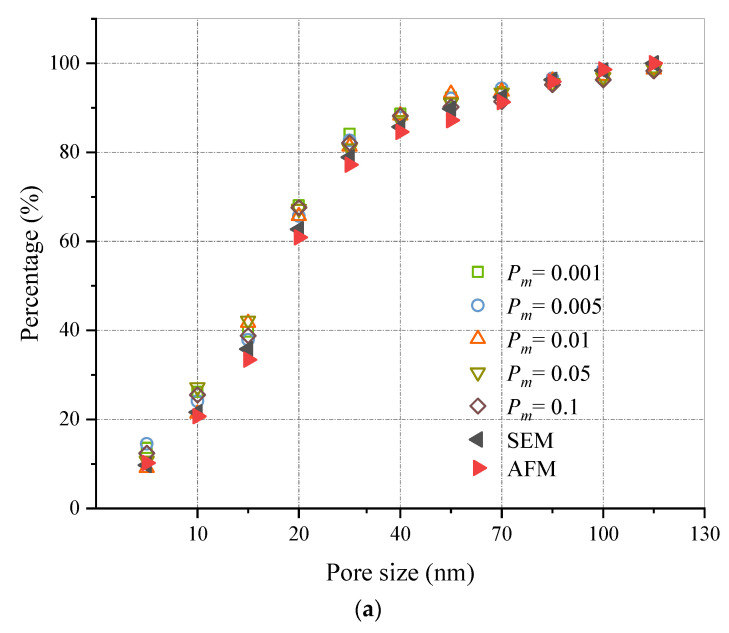
(**a**) Pore distribution statistics (*p_d_
*= 0.001). (**b**) Pore distribution statistics (*p_m_
*= 0.001).

**Figure 6 materials-15-08253-f006:**
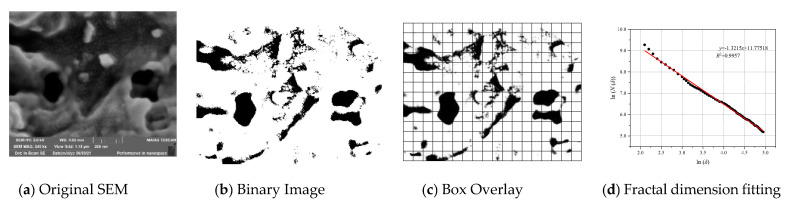
Schematic Diagram of Surface Fractal Dimension Calculation.

**Figure 7 materials-15-08253-f007:**
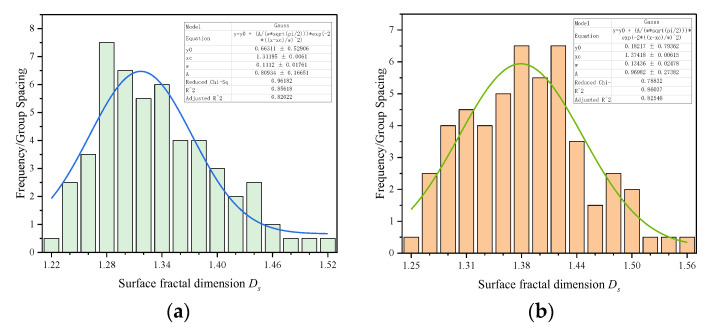
The distribution statistics of surface fractal dimension. (**a**) Surface fractal dimension *D_S_* of QSGS. (**b**) Surface fractal dimension *D_S_* of FE-SEM image.

**Figure 8 materials-15-08253-f008:**
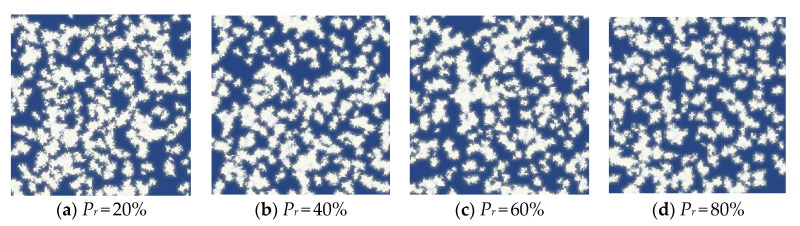
Calculation model of C-S-H.

**Figure 9 materials-15-08253-f009:**
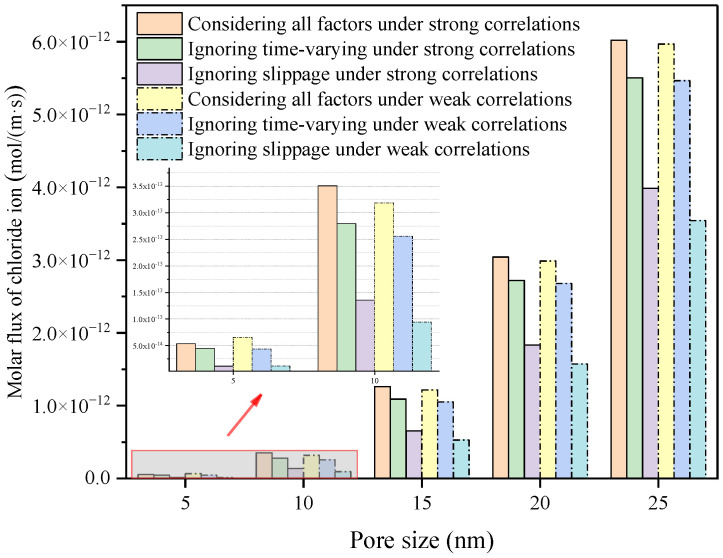
Molar flux of chloride ions in different pore sizes at steady state.

**Figure 10 materials-15-08253-f010:**
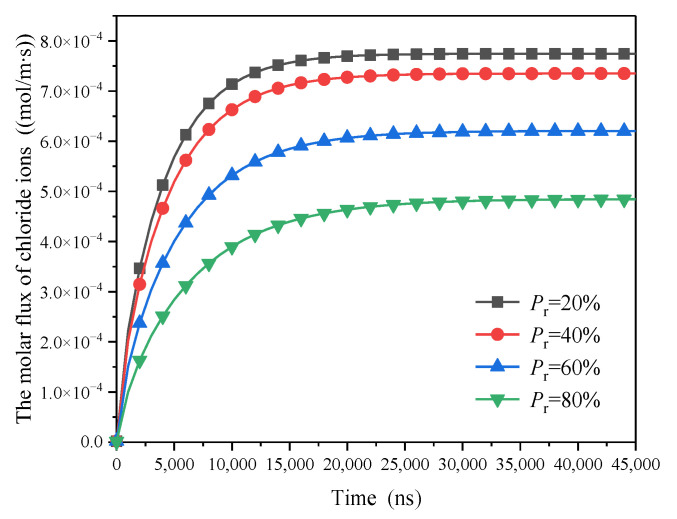
The influence of the ratio of HD C-S-H on molar flux of chloride ions.

**Figure 11 materials-15-08253-f011:**
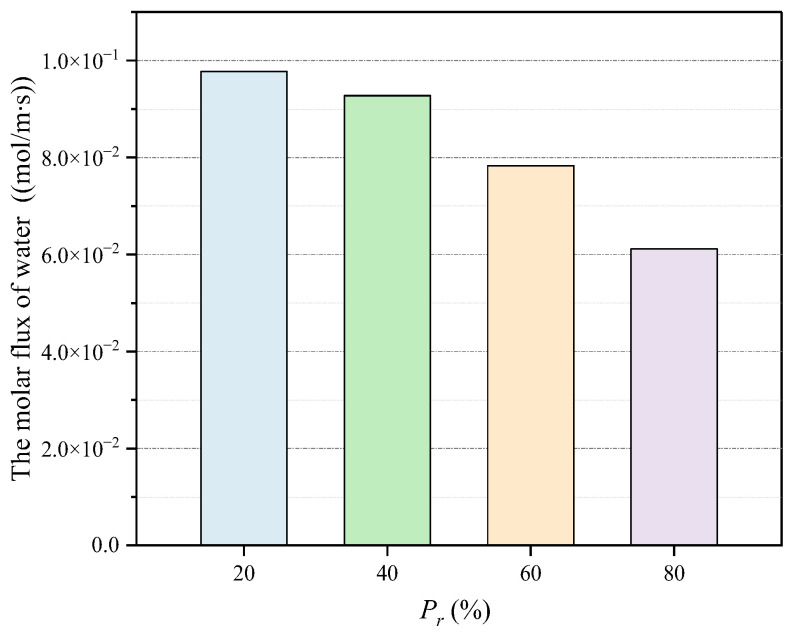
The influence of the ratio of HD C-S-H on molar flux of water at steady state.

**Figure 12 materials-15-08253-f012:**
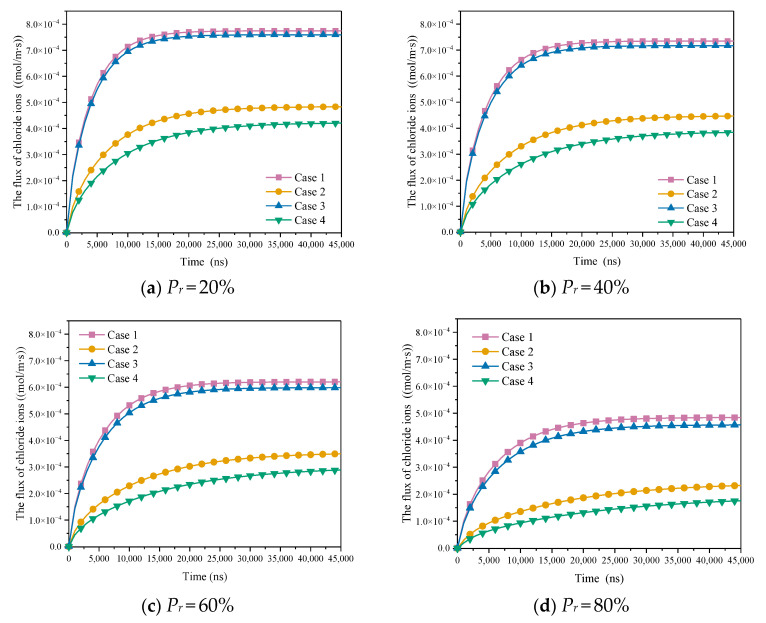
The influence of slippage to molar flux of chloride ions at micron scale.

**Figure 13 materials-15-08253-f013:**
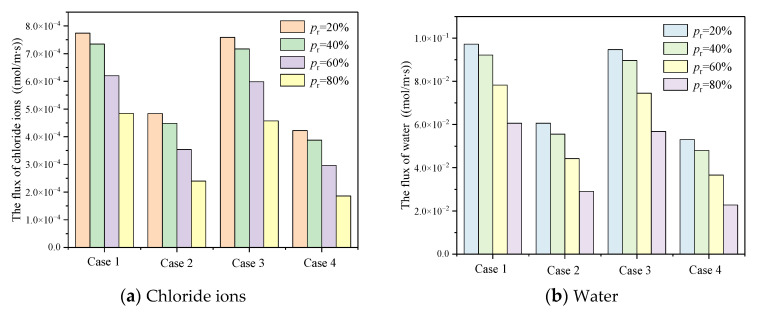
The influence of slippage on diffusion flux on steady state at micron scale.

**Figure 14 materials-15-08253-f014:**
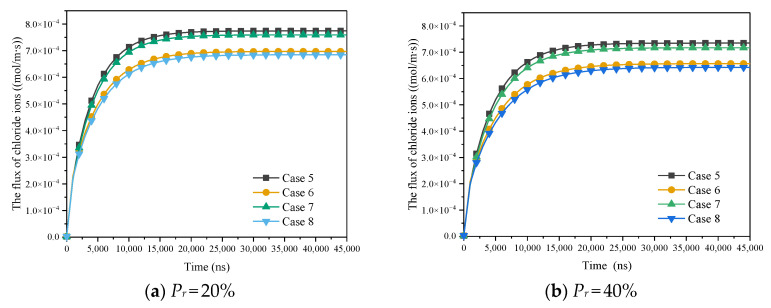
The influence of EDL time-varying on molar flux of chloride ions.

**Figure 15 materials-15-08253-f015:**
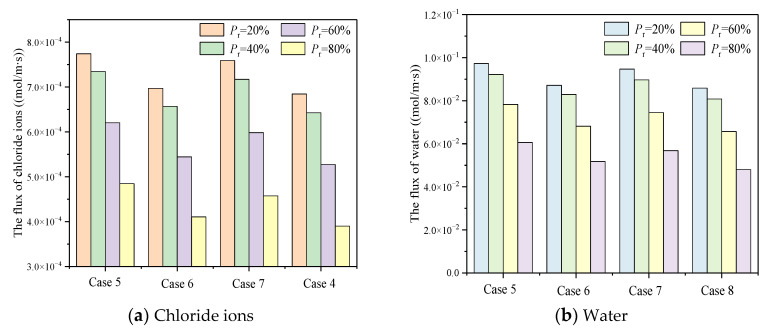
The influence of EDL time-varying on diffusion flux at steady state at micron scale.

**Figure 16 materials-15-08253-f016:**
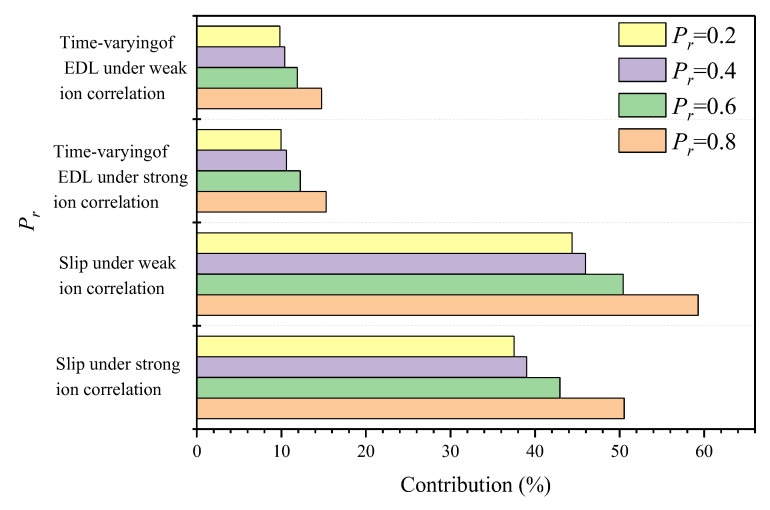
Contribution of interface effect under combined factors.

**Table 1 materials-15-08253-t001:** Observation and analysis results of nanopore in C-S-H (the analysis area is 50,000 ${\mu m}^{2}$).

Sample Number	Total Number of Pores	$\mathrm{Average}\;\mathrm{Pore}\;\mathrm{Area}\;(nm^{2})$	$\mathrm{Maximum}\;\mathrm{Pore}\;\mathrm{Area}\;(nm^{2})$	$\mathrm{Minimum}\;\mathrm{Pore}\;\mathrm{Area}\;(nm^{2})$	$\mathrm{Total}\;\mathrm{Pore}\;\mathrm{Area}\;(\mu m^{2})$
1	$1{.74\;\times \;10}^{6}$	649	38,421	163	1130
2	$2{.17\;\times \;10}^{6}$	502	31,668	54	1090
3	$3{.73\;\times \;10}^{5}$	225	13,266	21	83.9
4	$3{.75\;\times \;10}^{5}$	182	22,561	43	68.3

**Table 2 materials-15-08253-t002:** Observation and analysis results of nanopore in C-S-H (the analysis area is 0.01 mm^2^).

Sample Number	Total Number of Pores	$\mathrm{Average}\;\mathrm{Pore}\;\mathrm{Area}\;(nm^{2})$	$\mathrm{Maximum}\;\mathrm{Pore}\;\mathrm{Area}\;(nm^{2})$	$\mathrm{Minimum}\;\mathrm{Pore}\;\mathrm{Area}\;(nm^{2})$	$\mathrm{Total}\;\mathrm{Pore}\;\mathrm{Area}\;(\mu m^{2})$
1	$3{.67\;\times \;10}^{5}$	568	32,634	154	208.46
2	$4{.19\;\times \;10}^{5}$	474	29,840	38	198.61
3	$6{.74\;\times \;10}^{4}$	189	11,304	14	12.74
4	$7{.53\;\times \;10}^{4}$	165	21,371	29	12.43

**Table 3 materials-15-08253-t003:** The proportion of HD C-S-H in different cases.

Number	Case a	Case b	Case c	Case d
*P_r_ *(%)	20	40	60	80

**Table 4 materials-15-08253-t004:** Factors considered in different cases.

Number	Case 1	Case 2	Case 3	Case 4
Factor	Considering slip under strong ion correlation	Ignoring slip under strong ion correlation	Considering slip under weak ion correlation	Ignoring slip under weak ion correlation

**Table 5 materials-15-08253-t005:** Factors considered in different cases.

Number	Case 5	Case 6	Case 7	Case 8
Factor	Considering time-varying of EDL under strong ion correlation	Ignoring time-varying of EDL under strong ion correlation	Considering time-varying of EDL under weak ion correlation	Ignoring time-varying of EDL under weak ion correlation

## Data Availability

The data generated and/or analyzed during the current study are not publicly available for legal/ethical reasons but are available from the corresponding author on reasonable request.

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
