# Peer review of "A Cross-Scale Framework for Modelling Chloride Ions Diffusion in C-S-H: Combined Effects of Slip, Electric Double Layer and Ion Correlation"

_materials, 2022, doi:10.3390/ma15228253_

Round 1
Reviewer 1 Report
According to the abstract I suggested that it will be possible to provide more useful review. Unfortunately, the paper is not in my scope of knowledge.
Thus, I am providing the cosmetic remarks only.
Line 44. What is the meaning of ITZ?
Line 105. Stern layer (with capital)?
Figure 1. The model needs much more description.
Line 258. Not clear
Line 299. Not clear
Line 329. Not clear
Tables 4 and 5. Correct the numbers in subtitles.
Figure 14d. What is the axis-y?
Author Response
Thank you very much for your great help to improve this paper. Those comments are all valuable and very helpful for revising and improving our paper, as well as the important guiding significance to our researches. We have studied the comments carefully and have made corrections which we hope meet with approval. Please see the attachmentt for the specific revised contents.

Reviewer 2 Report
This study created a new cross-scale model covering slip effects, time-varying of EDL, and ion correlation. This work is very interesting and can provide new insights into improving the durability of concrete by changing the solid-liquid interface on the micro-nanoscale. However, some minor revisions should be done before this paper can be accepted.
1. The full name of an abbreviation should only be explained when it first appears in the article, and no further explanation is required when it appears again, for example, “electric double layer (EDL)” in line 93.
2. The English needs to be improved. Some sentences are too long and complicated. For example,Line 145-148, 192-197 and “It can be seen that the…" etc. I suggest the author can read and correct.
3. In line 306-308, the parameter Pr is affected by many factors such as water cement ratio and hydration time in the actual hydration process. The influence of these factors on the distribution of HD C-S-H and LD C-S-H should be properly explained. In addition, the author should also indicate the significance of discussing the proportion of HD C-S-H in practical engineering.
4. In line261-263, the surface fractal dimension of a single pore is calculated by box counting. We can estimate the authors' calculation methods to some extent, but it is not clear. The author should give a detailed calculation method of surface fractal dimension included in this paper
5. The discussion research work is deep. C-S-H is the main phase in cement pastes. The authors should describe the important role of C-S-H in dominating the strength and durability in the introduction part, thus the potential readers can understand better the importance of this work. Some related studies such as Comparison between the effects of phosphorous slag and fly ash on the C-S-H structure, long-term hydration heat and volume deformation of cement-based materials; Investigation of microstructure of C-S-H and micro-mechanics of cement pastes under NH4NO3 dissolution by 29Si MAS NMR and microhardness.
6. In lines 267-278, the fractal method was described. I think more related references could enrich this part and provide addition information about fractals, such as Comparison between the influence of finely ground phosphorous slag and fly ash on frost resistance, pore structures and fractal features of hydraulic concrete.
Overall, this is an interesting and excellent work, especially for those who study concrete durability. I recommend a minor revision before acceptance.
Author Response

(The authors gave the same response as above.)

Reviewer 3 Report
The manuscript is about the modelling chloride ions diffusion in C-S-H.
In general, the structure of the manuscript is well organized, the methods are well described, and the procedures are well conducted. The results sound and the conclusions are based on the experimental data.
Comments on the manuscript:
l.75-76: ‘Slip is a state of motion in which a part of a crystal moves along a certain crystal plane and direction relative to another part of the liquid film under the action of shear stress’. Crystals are solid structures with regular positions and distances of their constituents, therefore, the correct term would be ‘volume’.
l.215-216: Use µm instead of ‘microns’; min instead of ‘Min’; is the unit ‘nm Celsius’ correct?
The sample preparation was not described in the work: type of cement, water:cement ratio, mixing program, additives used, curing conditions, etc.
l.238: Use kV instead of ‘KV”.
l.315: Define LD C-S-H and HD C-S-H.
Fig.9: Error bars are missing, or, at least, the deviations of model should be given. The same for Fig.11.
Is it possible to give the uncertainty of the simulation? It would be interesting for practical analysis.
There are several references, but they are old! From 34 references, 23 are older than 2016! Also, some references are incomplete.
Author Response

(The authors gave the same response as above.)
